# PLANS: Neuro-Symbolic Program Learning from Videos

**Raphaël Dang-Nhu**
ETH Zürich
dangnhur@ethz.ch

## Abstract

Recent years have seen the rise of statistical program learning based on neural models as an alternative to traditional rule-based systems for programming by example. Rule-based approaches offer correctness guarantees in an unsupervised way as they inherently capture logical rules, while neural models are more realistically scalable to raw, high-dimensional input, and provide resistance to noisy I/O specifications. We introduce PLANS (Program LeArning from Neurally inferred Specifications), a hybrid model for program synthesis from visual observations that gets the best of both worlds, relying on (i) a neural architecture trained to extract abstract, high-level information from each raw individual input (ii) a rule-based system using the extracted information as I/O specifications to synthesize a program capturing the different observations. In order to address the key challenge of making PLANS resistant to noise in the network's output, we introduce a dynamic filtering algorithm for I/O specifications based on selective classification techniques. We obtain state-of-the-art performance at *program synthesis from diverse demonstration videos* in the Karel and ViZDoom environments, while requiring no ground-truth program for training.

## 1 Introduction

The problem of automatically generating a program satisfying given specifications is a long-standing challenge of artificial intelligence (Waldinger, Lee, 1969). The sub-area of *inductive program synthesis*, also known as *programming by example* (Gulwani, 2011), focuses on specifications formed of examples of desired I/O program behavior. In existing systems, I/O examples typically are instances of simple programming data types such as booleans, integers, floats or strings, or elementary data structures such as lists. Extending programming by example to more complex input domains is a very useful task, as it opens the range of possible real-world applications. Sun et al. (2018) made an important step in this direction by introducing the task of *program synthesis from diverse demonstrations videos*. This is a supervised learning problem in which the input is a set of videos of an agent's behavior provided as raw visual input, and the desired output is a program summarizing the decision making logic (or policy) of this agent. From the point of view of imitation learning, this task is important because policies represented as programs might generalize more strongly outside the training distribution, require less data to learn, and allow formal verification of safety properties.

Inductive synthesis techniques can be divided in two categories. The traditional *symbolic* or *rule-based* approaches (Jha et al., 2010; Alur et al., 2013) typically rely on efficient search through a user-defined domain specific language, leveraging constraint solvers based on SAT and SMT solving for search-space pruning. Alternatively, recent efforts aim at leveraging *statistical learning* methods, by phrasing inductive synthesis as a supervised learning task (Gaunt et al., 2016; Parisotto et al., 2017). Rule-based approaches have several advantages: they offer guarantees that the generated program is correct (i.e., satisfies the provided specifications). Additionally, they demonstrate better generalization to unseen inputs. The latter was experimentally shown by Gaunt et al. (2016) in the

Terpret framework, and confirmed by Devlin et al. (2017) on the FlashFill benchmark (Gulwani et al., 2012). However, they are computationally expensive and can suffer from scalability issues. As a result, they are inherently unable to handle raw visual input. Besides, they struggle to deal with incorrect input as they systematically attempt to satisfy all examples. Statistical approaches (see Allamanis et al. (2018) for a survey) appear as a way to mitigate these problems, by handling images (Ellis et al., 2018), scenes (Liu et al., 2019) or videos (Sun et al., 2018) as inputs, and learning to be robust against noise (Devlin et al., 2017). But this comes at the cost of generating large datasets labeled with ground-truth programs for training.

We introduce PLANS, a hybrid model for program synthesis from demonstration videos that combines neural and rule-based techniques in order to get the best of both worlds. It has two components: First, a neural architecture is trained to infer a high-level description of each individual behavior from raw visual observation. Second, a rule-based solver uses the information inferred by the network as I/O specifications to synthesize a program capturing the different behaviors. The neural component allows to handle raw visual input by transforming it into abstract specifications, while the rule-based system removes the need for ground-truth programs by inherently capturing logical rules. Therefore, our model uses strictly less supervision than prior work. We address the key challenge of making PLANS robust to the noise introduced by the imperfect inference of specifications, which has a dramatic impact on the solver's performance if not properly handled. Specifically, we introduce an adaptive filtering algorithm for I/O specifications based on selective classification techniques, in which only high-confidence predictions are passed to the solver. Using this method, PLANS clearly outperforms previous methods (Sun et al., 2018; Duan et al., 2019). Finally, our model maintains a reasonable temporal complexity, as it relies on several independent solver calls that can be performed in parallel. To summarize, we make the following contributions:

- We develop a neural architecture for inferring I/O specifications from videos and an encoding of the synthesis task into an off-the-shelf rule-based solver (Rosette (Torlak, Bodik, 2013)).

- We address the key problem of making the rule-based solver robust to noise in the network's output by developing an adaptive filtering algorithm for specifications.

- We evaluate PLANS on the Karel and ViZDoom benchmarks and show significant performance improvements compared to state-of-the-art end-to-end neural architectures, despite using strictly less supervision signals.

## 2 Background

### 2.1 Motivation

Understanding an agent's decision making logic is a long standing task in artificial intelligence, originally motivated by the performance benefits of imitation learning techniques (Schaal, 1997; Ng, Russell, 2000; Ross et al., 2011). In addition, it is related to making algorithms more interpretable to humans. In particular, recent work has focused on interpreting the neural policy representation in modern deep reinforcement learning algorithms (Lipton, 2018; Montavon et al., 2018). In this context, program representation of policies has several advantages. First, it gives a concise representation that is suitable for intuitive human understanding. Second, it permits to leverage symbolic verification techniques to ensure safety guarantees in the learning process (Verma et al., 2018; Zhu et al., 2019). The core aspect of these programmatic approaches is the ability to synthesize a *summarized* representation of the different observed behaviors of the agent.

The setting considered here is specific in two ways. First, most existing techniques assume that a high-level description of actions performed by the agent is always available. Second, the control-flow interpretability of the synthesized program is limited by the complexity of the underlying state space representation. This can lead to degenerate situations where the execution of the program depends on non-interpretable boolean conditions. For instance, in the case of visual input, this might be a threshold on the value of a single pixel in a $80 \times 80$ image. On the contrary, we assume that ground truth action sequences are not available at test time, and that the high dimension of the state-space requires restricting possible boolean conditions to high-level description of the state. In the context of playing video games, an example of meaningful condition is a boolean that indicates whether an enemy is present in the environment. These assumptions create a black-box setting where the specifications for synthesis of a program policy are not immediately available and need to be inferred.

## 2.2 Task description

Here, we describe the task of *program synthesis from diverse demonstration videos* (Sun et al., 2018).

**Agent-environment interaction**    We consider an environment with observable state $s \in \mathcal{S}$, and an agent interacting with this environment in discrete timesteps. We assume that the agent has a fixed set of possible actions $\mathcal{A}$, and that the influence of each action on the state of the environment is determined by a deterministic transition function $\mathcal{T} : \mathcal{S} \times \mathcal{A} \rightarrow \mathcal{S}$. Additionally, we suppose that the agent behaves according to an unknown, deterministic policy $\eta(s) \in \mathcal{A}$ that determines the next action depending on the sequence of previously observed states $s = (s_1, \ldots, s_i)$. Finally, we assume that the agent has control over the end of the interaction by means of a specific action $\mathrm{end} \in \mathcal{A}$.

**Demonstrations**    A demonstration $\tau$ is formed of a sequence of states $s = (s_1, \ldots, s_T)$ together with a sequence of actions of same length $a = (a_1, \ldots, a_T)$, satisfying the following conditions (i) The actions of the agent follow the policy $\eta$. That is, for all $i$ in $\{1 \ldots T\}$, we have $a_i = \eta((s_1, \ldots, s_i))$. (ii) The environment transitions are determined by $\mathcal{T}$, i.e. for all $i$ in $\{1 \ldots T - 1\}$, we have $s_{i+1} = \mathcal{T}(s_i, a_i)$. (iii) The sequence is complete, i.e. the last action $a_T$ is end. Since the transition function and policy are deterministic, the demonstration is uniquely determined by the environment's initial state $s_1$. Therefore, we can generate demonstrations for a policy by sampling random initial states and applying the policy and transition iteratively. In the case of visual state observations, the state sequence $s$ can be seen as video composed of successive frames.

**Perceptions**    In order to address the interpretability issue arising from the high dimensionality of $\mathcal{S}$, we assume the existence of $d$ perception primitives $\gamma_1, \ldots, \gamma_d : \mathcal{S} \rightarrow \{\mathrm{true}, \mathrm{false}\}$. Intuitively, the perceptions $(\gamma_1(s), \ldots, \gamma_d(s))$ give a high-level, abstract representation of the state. For instance, in the context of video games, $\gamma_1(s)$ could be true if and only if $s$ the agent is facing an enemy. The underlying idea is that the agent's behavior $\eta$ should only depend on the compact state representation provided by the perception primitives. Given a demonstration $\tau$, we define $d$ associated perception sequences $p^1, \ldots, p^d$ as $p^j = (p_1^j, \ldots, p_T^j) = (\gamma_j(s_1), \ldots, \gamma_j(s_T))$.

**Program synthesis from demonstration videos**    Sun et al. (2018) proposed the following supervised learning task, where the input is a set of demonstrations generated with an unknown policy $\eta$ and different initial states, and the desired output is a representation of $\eta$. The benchmark datasets are generated such that $\eta$ can always be represented by a program in a given domain specific language (DSL). The execution of the program specifies which sequence of actions is executed, with different control-flow constructs, such as conditional branchings and loops. The boolean conditions used for these constructs are only allowed to depend on the high-level representation $\gamma(s) = (\gamma_1(s), \ldots, \gamma_l(s))$ of the state obtained via the perception primitives. The DSL is formally defined in Figure 1.

**Supervision signals**    Sun et al. (2018) make the assumption that neither the action nor the perception sequences are available at test time. They can only be used as a supervision signal in the training phase. PLANS conforms to this assumption, explaining the need for a neural model that learns to infer the I/O specifications. Compared to previous work, the main specificity of PLANS is that it does not require program supervision. That is, the ground-truth program is never accessible to the model.

## 2.3 Summary of previous approaches

**demo2program**    Sun et al. (2018) designed an end-to-end neural model for the task of program synthesis from demonstration videos. It is based on the following components (i) a convolutional neural network to encode each of the video frames (ii) an *encoder* LSTM layer that takes as input each video as a sequence of frame encodings (iii) a *summarizer* LSTM layer that re-encodes the video with aggregated video encodings as initial state (iv) a relational network module that summarizes the different video encodings (v) a final *decoder* LSTM layer that outputs the program in the form of a sequence of code tokens. They demonstrate that the summarizer LSTM layer as well as the relational module are essential to obtain reliable identification of the correct program control-flow.

**watch-reason-code**    Duan et al. (2019) proposed to reduce the memory footprint of the architecture by introducing a deviation-pooling strategy replacing the relational module, and to use multiple decoding layers to refine the generated program, obtaining slight performance improvements.

# 3 The PLANS model

Figure 2 gives a high-level overview of PLANS. We describe the neural component (Section 3.1), the rule-based solver (Section 3.2) and our adaptive filtering algorithm (Section 3.3).

## 3.1 Neural inference of specifications

The first component of PLANS is a neural architecture that learns to infer the action and perception sequences from raw visual input. This formalizes as a sequence-to-sequence learning task where the input is a video demonstration $\tau = (s_1, \ldots, s_T)$, and the desired outputs are (i) the ground-truth action sequence $(a_1, \ldots, a_T)$ (ii) the ground-truth perception sequences $(p_1^1, \ldots, p_T^1) \ldots (p_1^l, \ldots, p_T^l)$. To perform this task, we used a vanilla seq2seq model enhanced with an attention mechanism. In order to interpret visual input, we use a multi-layer convolutional network as a state embedding, i.e the state $s_i$ is encoded by a convolutional network to a state vector $v_i$ before it is fed to the encoder layer. We use two different decoder layers for actions and perceptions, while the embedding and encoder layer are common for both action and perceptions. To ensure reproducibility of our results, extensive description of hyperparameters and training process can be found in the supplementary material.

## 3.2 Encoding into an inductive synthesis problem

Our rule-based synthesizer is implemented in Rosette (Torlak, Bodik, 2013), which itself builds on the Z3 solver (De Moura, Bjørner, 2008). Rosette allows to specify a program with holes (or *sketch* (Lezama, 2008)), and a desired behavior. Then, it automatically fills the holes by encoding the resulting problem into a set of logical constraints that have to be satisfied. A very convenient functionality of Rosette is the possibility to specify complex holes that can be filled with expressions from a predefined grammar, allowing to encode the DSL of Figure 1, and restricting the search to valid programs. In the supplementary material, we show how we encoded the different control-flow constructs of the DSL in Rosette.

$$
\begin{array}{lll}
Program\ m & \to & s\ ;\ \text{end} \\
Statement\ s & \to & s_1\ ;\ s_2 \\
& | & \text{while}(b) : s \\
& | & \text{repeat}(r) : s \qquad r \in \mathbb{N} \\
& | & \text{if}(b) : s\ \text{else} : s2 \\
& | & \text{if}(b) : s \\
& | & Action\ a \qquad\qquad a \in \mathcal{A} \\
Condition\ b & \to & \text{not}\ b \\
& | & Perception\ \gamma_i \qquad i \in \{1 \ldots l\}
\end{array}
$$

Figure 1: DSL for representing policies.

The task of program synthesis from diverse demonstration videos naturally phrases as a programming by example problem. The I/O specifications contain the perception sequences, which serve as program input, and the corresponding action sequence, which is the desired output. In order to achieve good generalization of the synthesized program to unseen demonstrations, it is crucial to privilege simpler solutions among several programs satisfying the I/O specifications. That is, we aim at finding the solution with the least cost, defined as the number of control-flow branchings (if or while statements). Since Rosette has no native support for cost functions, we propose to bound the number of control-flow statements in the Rosette encoding of the DSL, and make repeated calls to the solver while increasing this bound, until the resulting set of constraints is satisfiable. We give additional details about this procedure in the supplementary material.

Rule-based synthesizers have the advantage that they are both *sound* and *complete*: any program returned by the solver is guaranteed to satisfy all the provided specifications, and if there exists a satisfying way to fill the holes, the solver will find a solution in finite time (in the case of bounded number of control-flow statements). These guarantees are usually associated with a high computational cost. In the case at hand, thanks to the relative simplicity of the DSL which contains no variable assignments, we did not observe time to be a problem. **All solver queries terminate in a few seconds**, which in orders of magnitude is similar to the time needed by the neural architecture to infer the different specifications. Besides, the different calls to the solver are independent and can be made in parallel. We give precise time measurements in the supplementary material.

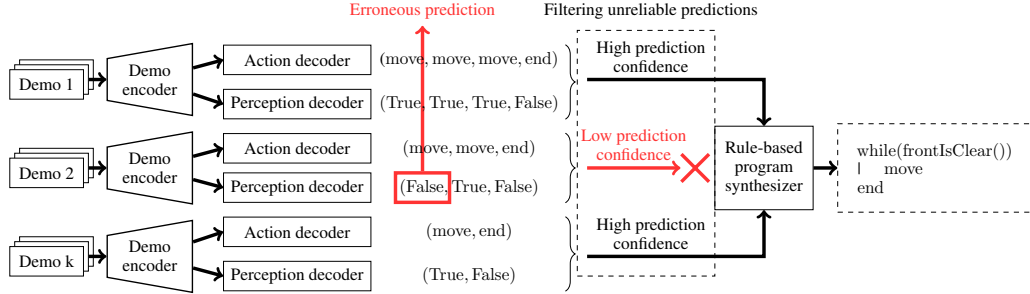

Figure 2: Overview of PLANS. The video demonstrations are individually fed at pixel-level into the neural encoder-decoder architecture, which infers the high-level action and perception sequences. For the sake of clarity, we only show one perception sequence corresponding to the $\mathrm{frontIsClear}()$ perception primitive. The different sequences are then given to a rule-based solver, that generates a program summarizing the different demonstrations. The demonstration with inconsistent perception sequence is identified thanks to its low prediction confidence, and is not provided to the solver.

## 3.3 Synthesis from noisy specifications

The ability to generate correct perception-action specifications for the synthesis task is crucial to the performance of PLANS. Indeed, existing rule-based solvers are very sensitive to even small amounts of input noise (Devlin et al., 2017). Consequently, our model can only achieve satisfying performance if all the specifications given to the solver are reliable. While this assumption is met on the Karel benchmark, we observe in the ViZDoom environment that the quasi totality (above 99%) of test programs have at least one erroneous predicted action or perception token among the 25 observed demonstrations. Existing work on rule-based synthesis from noisy data (Raychev et al., 2016) propose dataset sub-sampling as a way to avoid overfitting to incorrect specifications. This is not directly applicable to our setting. Indeed, it only guarantees to find a program that is close to the correct solution, in the sense that it satisfies as many I/O specifications as possible, with a focus on datasets containing a large number of examples. In contrast, we deal with a fixed and limited amount of demonstrations, and aim at identifying exactly the original program.

Instead, we designed a simple yet effective filtering algorithm for demonstrations, based on the principle of selective classification. This method makes use of the neural network's prediction confidence. For a given sequence token, the prediction confidence is defined as the probability of the predicted class in the softmax output layer. The filtering works in two steps: First, we assign to each demonstration $\tau$ an *action confidence level* $\mathrm{actionconf}(\tau) \in [0, 1]$ and a *perception confidence level* $\mathrm{perconf}(\tau) \in [0, 1]$ to characterize the global confidence of the whole sequence of predictions. Then, if either one of the two confidence levels is too low, we discard the whole demonstration. We explore two variants of this method: a static filtering method that uses fixed thresholds for the action and prediction confidence, and a dynamic method that incrementally increases the thresholds while making repeated calls to the solver, until a valid solution is found. Before we describe these algorithms in detail, let us formally define the action and prediction confidence levels.

**Definition.** *Let $\tau$ be a demonstration, for which the neural architecture predicts action sequence $a = (a_1, \ldots, a_T)$ and perception sequences $p^1 = (p_1^1, \ldots, p_T^1), \ldots, p^l = (p_1^l, \ldots, p_T^l)$. Let us denote $\mathrm{conf}(a_i)$ (resp $\mathrm{conf}(p_i^j)$) for the prediction confidence of action token $a_i$ (resp perception token $p_i^j$), which is defined as the probability of $a_i$ (resp. $p_i^j$) in the softmax output layer. We define the action confidence $\mathrm{actionconf}(\tau)$ of the demonstration as the minimum of the action tokens confidence. That is,*

$$\mathrm{actionconf}(\tau) = \min_i \mathrm{conf}(a_i).$$

*Similarly, we define the perception confidence $\mathrm{perconf}(a)$ of $\tau$ to be the minimum of the perception tokens confidence, formally*

$$\mathrm{perconf}(\tau) = \min_{i,j} \mathrm{conf}(p_i^j).$$

*Note that for the perception confidence level, the min is taken both across timesteps (indexed by $i$) and perception primitives (indexed by $j$).*

**Static filtering** In the static filtering strategy, we fix two thresholds $\epsilon_a$ and $\epsilon_p$ for the action and perception confidence respectively. Then, we filter all the I/O examples for which the underlying demonstration $\tau$ verifies $\mathrm{actionconf}(\tau) < \epsilon_a$ or $\mathrm{perconf}(\tau) < \epsilon_p$. $\epsilon_a$ are $\epsilon_p$ are treated as hyper-parameters and optimized on the validation dataset. Tuning the thresholds creates a trade-off between the number of remaining demonstrations and their reliability.

**Dynamic filtering** In practice, we observe the static filtering strategy to be efficient at filtering out incorrect action sequences. However, it is sub-optimal for filtering perception sequences, because the best threshold $\epsilon_p$ varies significantly across programs. To mitigate this problem, we employ a dynamic filtering strategy for perception sequences. We sort the demonstrations by increasing perception confidence, i.e. we have $\tau_1, \ldots, \tau_k$ with $\mathrm{perconf}(\tau_1) < \ldots . \mathrm{perconf}(\tau_k)$. Then, we incrementally filter out the demonstrations by increasing order of confidence, while making repeated calls to the program synthesizer, until the resulting set of constraints is satisfiable. This progressively reduces the number of I/O examples, with the hope that incorrect predictions will have low confidence and will be filtered first. This dynamic filtering strategy is adaptive in the sense that it removes the need for tuning the threshold $\epsilon_p$. It is coupled with the search for optimal program cost in the following way: in the outer loop, we increment the perception threshold $\epsilon_p$, and in the inner loop, we increment the number of control-flow statements. In the supplementary material, we provide a detailed description of this algorithm. We also experimented with a decremental scheme and obtained similar accuracy.

# 4   Experimental results

In this section, we report experimental observations concerning the performance of PLANS.

**Benchmarks** Karel (Pattis, 1981) is an educational programming language that controls a robot navigating through a grid world with walls and markers. ViZDoom (Kempka et al., 2016) is an open-source platform for Doom, a classical first-person shooter game. It allows training bots via reinforcement learning from visual observations. We use the three evaluation metrics designed by Sun et al. (2018). To measure *execution accuracy*, we compare if the predicted and ground-truth programs behave similarly on a fixed number of not previously observed initial states. *Sequence accuracy* measures if the predicted and ground-truth programs match exactly. *Program accuracy* is similar to sequence accuracy, with identification of some semantically equivalent programs: e.g., $\mathrm{repeat}(3) : \mathrm{move}$ and $\mathrm{move} : \mathrm{move} : \mathrm{move}$ will be considered as equivalent by program accuracy.

**Experimental setup** We performed all experiments on a machine with 2.00GHz Intel Xeon E5-2650 CPUs and using a single GeForce RTX 2080 Ti GPU. We make our implementation public and provide additional details about the experiments duration in the supplementary material.

## 4.1   Comparison with demo2program and watch-reason-code

Results on the main Karel and ViZDoom benchmarks are presented in Table 1. Values for both baselines are directly reported from prior work, and show best obtained performance. For PLANS, we report mean and standard deviation over three independent runs. PLANS strongly improves execution accuracy compared to prior work: approximately $+15\%$ absolute improvement for Karel, and $+10\%$ for ViZDoom. This means that PLANS is significantly better at capturing the different possible behaviors. We also observe improvements of program accuracy, though less significant. We surmise that this is due to imperfections of the program accuracy metric, which captures some but not all semantically equivalent programs. For instance, we observe that $\mathrm{if}(\mathrm{frontIsClear}()) : \mathrm{move} \ \mathrm{else} : \mathrm{turn}$ and $\mathrm{if}(\mathrm{not} \ \mathrm{frontIsClear}()) : \mathrm{turn} \ \mathrm{else} : \mathrm{move}$ are not recognized as equivalent. An important direction for future work is to improve this metric in the original benchmark released by Sun et al. (2018). Finally, we observe no sequence accuracy improvement in the Karel benchmark. However, we believe that the obtained performance is still very reasonable. Indeed, as opposed to the two baselines, PLANS does not access any ground-truth program during training. Therefore, it can not be expected to distinguish between semantically equivalent programs.

In the ViZDoom environement, our results confirm that the performance of PLANS heavily relies on the filtering heuristics. Without filtering, the model achieves very poor performance. With static filtering only, the model achieves reasonable accuracy but fails to outperform both baselines. Dynamic filtering yields the best results.

Table 1: Accuracy comparison on the main Karel and ViZDoom benchmarks. Values are in %.

| % | Karel | | | ViZDoom | | |
|---|---|---|---|---|---|---|
| Model | Execution | Program | Sequence | Execution | Program | Sequence |
| demo2program | 72.1 | 48.9 | 41.0 | 78.4 | 62.5 | 53.2 |
| watch-reason-code | 74.7 | 51.2 | **43.3** | 68.1 | 63.4 | 55.8 |
| PLANS (none) | **91.6 ± 1.3** | **53.9 ± 1.0** | 34.2 ± 0.5 | 25.8 ± 0.9 | 21.7 ± 0.6 | 19.3 ± 0.6 |
| PLANS (static) | | | | 77.5 ± 1.5 | 57.9 ± 0.9 | 51.2 ± 0.8 |
| PLANS (dynamic) | | | | **88.0 ± 0.6** | **65.5 ± 0.6** | **58.8 ± 0.6** |

Table 2: Comparison on the ViZDoom if-else benchmark.

| % | ViZDoom if-else | | |
|---|---|---|---|
| Model | Execution | Program | Sequence |
| demo2program | 89.4 | 69.1 | 58.8 |
| watch-reason-code | 82.1 | 67.8 | 57.7 |
| PLANS (none) | 16.3 ± 1.2 | 13.8 ± 1.0 | 10.2 ± 0.6 |
| PLANS (static) | 77.0 ± 0.6 | 62.4 ± 0.7 | 50.2 ± 0.7 |
| PLANS (dynamic) | **91.4 ± 0.7** | **74.6 ± 0.7** | **62.1 ± 0.7** |

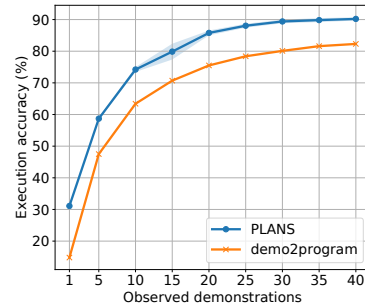

Figure 3: Influence of number of observed demonstrations on exec. accuracy (ViZDoom, main benchmark).

## 4.2 Additional experiments

**Number of observed demonstrations** In Figure 3, we show the influence of the number of observed demonstrations on ViZDoom execution accuracy. We observe consistent improvement over demo2program when the number of demonstrations increases. We do not report values for watch-reason-code as it is not given by prior work and no implementation is provided. Watch-reason-code is anyway significantly less accurate than demo2program for this metric. This experiment confirms the claim that our model reliably yields superior performance on the ViZDoom benchmark, and generalizes better to unseen initial states.

**If-else dataset** In order to specifically assess the ability of PLANS to identify control-flow divergences among demonstrations, we consider a specific dataset with programs composed of only one if-else branching statement. Results are shown in Table 2. Contrary to the main experiment, we observe similar improvements on all three metrics. We attribute this observation to the fact that programs in this dataset have less semantically equivalent variations, because of their simple form.

## 5 Related work

**Learning graphics programs from images and scenes** A related line of work is focused on inference of graphical programs from visual input (Ellis et al., 2015; Ganin et al., 2018; Liu et al., 2019; Tian et al., 2019). Most notably, Ellis et al. (2018) also use a hybrid architecture composed of a neural network and a rule-based solver. However, it can not be considered a programming by example task, as the specification only relates to a single input image, for which a compact, synthetic representation is desired. In contrast, we address the challenging task of summarizing the decision making in several videos, which requires to identify the control-flow divergences between the different observed behaviors. Besides, they ensure robustness against noise by incrementally adding specifications while rendering the resulting image and measuring its similarity with the input. This approach to identifying wrong specifications is task-specific as it requires the ability to incrementally render the generated specifications into an image, and does not apply to our setting.

**Synthesis of programs from noisy examples**   Our work confirms the idea that naive rule-based solvers are very vulnerable to noise in specifications (RobustFill, (Devlin et al., 2017)). We propose a general approach to mitigate this problem when the I/O specifications are inferred by a statistical learning system, and the noise originates from inaccuracies of this system. We believe that such settings will become increasingly common with the rise of hybrid neuro-symbolic models for machine reasoning. According to the taxonomy of Raychev et al. (2016), our filtering algorithms fall into the category of *dataset cleaning* methods, aiming at removing incorrect specifications before they are fed to the solver. Other categories include (i) probabilistic program synthesis (Nori et al., 2015), which relaxes the specifications into stochastic constraints (ii) genetic programming as a way of exploring large solution spaces while maximizing an objective function (Cramer, 1985; Baker, 1987). We do not compare with the algorithm of Raychev et al. (2016) because it does not build easily on off-the-shelf solvers, and requires unrealistic re-implementation.

**Neurally enhanced program synthesis**   An important line of work aims at using neural architectures to speed-up rule-based program synthesis by guiding the search process. DeepCoder (Balog et al., 2017) predicts an order on the program space in order to guide the rule-based solver towards potential solutions. Lee et al. (2018) propose to learn a probabilistic model attributing a likelihood to each program, with the aim of first exploring likely solutions. Ellis et al. (2018) learn a bias optimal strategy based on Levin search (Levin, 1973), with the goal of efficiently allocating compute time to different parts of the program search space. We did not need to use such approaches as synthesis was reasonably fast in the Karel and ViZDoom benchmarks.

**Programmatic reinforcement learning**   The idea of representing reinforcement learning policies as programs is investigated in a growing body of work. However, the search space of programs is highly non-smooth and makes the resulting problem intractable. Different works aim at mitigating this problem by first training a good neural policy via deep reinforcement learning. The policy is then used as an oracle providing I/O examples to generate a programmatic policy with similar behavior (Bastani et al., 2018; Verma et al., 2018, 2019). Despite some high-level similarities, the setting of program synthesis from diverse demonstration videos has some fundamental particularities (i) we aim at finding at program matching exactly the different examples, rather than just exhibiting similar behavior (ii) we consider that high-level description of each video is only available at training time (iii) we only dispose of a fixed, limited number of demonstrations, while the neural policy can be used to generate an arbitrary number of examples.

**Black-box imitation learning**   There exists prior work in the field of imitation learning which does not assume access to agent's actions. Torabi et al. (2018) develop a method for behavioral cloning from state observations. In (Nair et al., 2017), a robot learns pixel-level inverse dynamics by examining the influence of its own actions, in order to be able to infer the actions of the expert. However, this line of work does not consider program representations of policies.

**Selective prediction**   Our filtering strategies relates to the concept of selective prediction in statistical learning, also known as reject option. Since the seminal work of Chow (1957), the idea of rejecting certain predictions because of their low confidence has been extensively studied in various settings (Fumera, Roli, 2002; Geifman, El-Yaniv, 2017). Our action and prediction confidences can be seen as extensions of the softmax response (Geifman, El-Yaniv, 2019) for sequence prediction tasks. Other works have proposed to integrate the reject option in the learning process Cortes et al. (2016): a potential direction for future work is to integrate these mechanisms in our neural architecture. In the specific field of statistical program learning, very recent work has proposed to use selective prediction in order to ensure adversarial robustness for code (Bielik, Vechev, 2020).

## 6   Conclusion

A crucial challenge for intelligent systems is the ability to perform abstract reasoning from raw, unstructured data. To achieve this goal, prior work (Ellis et al., 2018) has evidenced the power of combining formal rule-based techniques with neural architectures. PLANS is the first application of such hybrid systems to the challenging task of identifying an agent's decision making logic from multiple videos. It sets up a new state-of-the-art for this task, with strictly less supervision signals. To efficiently combine neural and rule-based components, we developed an adaptive filtering algorithm

for neurally inferred specifications. We believe that this method is quite general and will be applicable to similar hybrid neuro-symbolic systems. For future work, we also aim at developing fine-grained filtering algorithms that only remove tokens instead of full sequences. This might help improving further the sample efficiency of our method.

## Acknowledgments

We thank Frederik Benzing and Wouter Tonnon for helpful discussions and proofreading.

## Funding disclosure

The author was funded by ETH Master Scholarship Programme during his studies.

## Broader Impact

The ability to understand agents' decision making logic from visual input can be applied to real-world observations such as videos of people driving. In this context, a simple example of logic that can be represented by a program is: "if the traffic light is green, move, else stop". Since PLANS requires no program supervision, it extends the applicability of prior work to new settings where no ground-truth programs are available. We believe that PLANS can eventually be applied in all contexts involving interaction between agents (human, animal or robotic) and their environment.

In the case of human agents, it is essential to foresee the impact on privacy of the observed subjects. In order to avoid privacy violations, it must be carefully assessed in each application of PLANS which components of a person's decision making logic can be interpreted without being intrusive. While the concern of data privacy is not specific to PLANS, it opens up new ethical and legal questions about the exact meaning and status of describing an agent's behavior.

It is also crucial to protect individuals from misinterpretation of their behavior. Prior systems for program synthesis from visual observations essentially have two failures modes: The first one corresponds to the case where the behavior in one individual video is not understood correctly, which will lead the synthesized program to capture behaviors that have not been actually observed. The second one occurs when the different videos are individually correctly interpreted, but the branching between the different behaviors (represented by the program control-flow) is not correctly inferred. PLANS addresses the first problem by leveraging the prediction confidence of its neural component, and the second by using a rule-based system that offers correctness guarantees. These aspects make PLANS more robust than prior work, which implies more reliability for end-users.

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
