[Supplementary Material]

# Supplementary Material of PLANS: Neuro-Symbolic Program Learning from Videos

**Raphaël Dang-Nhu**
ETH Zürich
`dangnhur@ethz.ch`

We provide the following appendices

- In A, we give additional information about the datasets (Karel and ViZDoom).
- In B, we describe precisely the neural component of PLANS and its training process.
- In C, we present the implementation of the rule-based solver.
- In D, we analyse the temporal complexity of PLANS.

## A Datasets

Table 3 contains high-level information about both datasets. We point to the following three differences that are relevant to our experimental results:

- Contrary to Karel, the ViZDoom demonstrations have a first-person point of view. Because of this subjective view, the environment is sometimes only partially observable, for instance when the agent's field of vision is occluded by a monster. In this situation, there might be a doubt about whether a second monster is hidden behind the first one. This accounts for the high number of uncertain predictions of actions and perceptions in ViZDoom.
- The resolution of observations is significantly larger in the ViZDoom environment, and there are more possible actions. This explains the need for a longer training of the neural component in the ViZDoom environment.
- In the ViZDoom environment, PLANS has access to more demonstrations to infer the underlying program. This is why we achieve similar accuracy on Karel and ViZDoom, despite the difficulties mentioned above.

For additional information about the datasets, we refer to Sun et al. (2018). Figures 4 and 5 give examples of programs and demonstrations for the Karel and ViZdoom benchmarks respectively. These images are taken from Sun et al. (2018). Because of the slightly different formalism in this paper, the end action is made implicit. The exact list of action and perception primitives is accessible in the original demo2program [1] repository.

## B Architecture and training

All video frames are first encoded with a convolutional neural network. All convolutional layers have kernel size 3 and stride 2. For the Karel dataset, we use 3 layers with respectively 16, 32 and 48 channels. For The ViZDoom dataset, we use 5 layers with respectively 16,32,48,48 and 48 channels. All the layers have LeakyRELU (Maas et al., 2013) activation and batch normalization. The resulting frame encodings are fed to a LSTM layer with 512 hidden units. We use two different LSTM layers for decoding: one predicts the action sequence, the other the perception sequences. Both decoding

Table 3: Dataset properties.

s

| Environment | Karel | ViZDoom |
|---|---|---|
| Point of view | Third person | First person |
| Frame resolution | 8x8 | 80x80 |
| # Actions | 5 | 11 |
| # Perceptions | 5 | 6 |
| # Observed demonstrations | 10 | 25 |
| # Unseen demonstrations | 5 | 5 |
| # Training samples | 30000 | 80000 |
| # Test samples | 5000 | 8000 |
| Max program size | 43 | 32 |

```
def run():
    while frontIsClear():
        move()
    putMarker()
    turnLeft()
    move()
    putMarker()
    move()
    move()
```

Figure 4: Example program and demonstrations for the Karel dataset. Figure from Sun et al. (2018).

layers have 512 hidden units. For both action and perceptions, we use a softmax output layer for predicting the probability of all classes. This encoder-decoder model is enhanced with the attention mechanism from Luong et al. (2015).

All models are trained with the Adam optimizer, using default parameters and learning rate (0.001). In Table 4, we give batch size and number of training steps for both our model and the demo2program

**Demo 1**

inTarget HellKnight

inTarget HellKnight
→ attack()

not inTarget Demon
→ moveRight()

**Demo 2**

inTarget HellKnight and
inTarget Demon

inTarget HellKnight
→ attack()

inTarget Demon
→ attack()

**Underlying Program**

```
def run():
    if inTarget HellKnight:
        attack()
    if not inTarget Demon:
        moveRight()
    else:
        attack()
```

Figure 5: Example program and demonstrations for ViZDoom. Figure from Sun et al. (2018).

Table 4: Number of training steps.

| Model | Karel | | ViZDoom | | | |
| | | | Phase 1 | | Phase 2 | |
| | Steps | Batch size | Steps | Batch size | Steps | Batch size |
|---|---|---|---|---|---|---|
| demo2program | ? | 128 | 50000 | 32 | 50000 | 8 |
| PLANS | 10000 | 32 | 30000 | 8 | | |

baseline. To the best of our knowledge, the number of training steps for demo2program on the Karel environment has not been provided by Sun et al. (2018).

## C  Details about solver implementation

In C.1, we present two algorithms that describe the exact order of solver calls with static and dynamic filtering respectively. In C.2, we detail which heuristics were used to improve the program and sequence accuracy metric. In C.3, we show how the different control-flow constructs were encoded in the Rosette solver.

### C.1  Algorithms

Here, we describe formally the static and dynamic filtering algorithms (Algorithms 1 and 2). The confidence thresholds $\epsilon_a = 0.98$ and $\epsilon_p = 0.9$ were chosen to yield best performance on the validation dataset. For the dynamic filtering heuristic, the perception confidence threshold is incremented such that we keep a fixed proportion prop of the demonstrations. We use a sequence of 11 predetermined proportions: $[1, 0.95, 0.9, 0.8, 0.7, 0.6, 0.5, 0.4, 0.3, 0.2, 0.1]$. The optimal sequence length is a trade-off between performance and number of solver calls. It is fine to increase the number of solver calls to a certain extent as they are independent and can be performed in parallel. However, efficient parallelism will require a higher number of cores.

---

Algorithm 1: Calls to the solver, with static filtering.

---

**Input:** Set of demonstrations $\mathcal{T}$ with neurally inferred action and perception sequences
**Output:** Program summarizing the different demonstrations, or unsat

$\quad \mathcal{G} \leftarrow \{\tau \in \mathcal{T} \mid \text{actionconf}(\tau) \geq 0.98 \wedge \text{perconf}(\tau) \geq 0.9\}$ $\qquad \triangleright$ Static filtering
$\quad$ **for** $n \in \text{range}(\text{max\_n})$ **do** $\qquad \triangleright$ Progressively increase number of control-flow statements
$\qquad$ solution $\leftarrow \text{solver}(\mathcal{G}, n)$ $\qquad \triangleright$ Call to Rosette solver
$\qquad$ **if** solution is not unsat **then return** solution
$\quad$ **return** unsat

---

---

Algorithm 2: Calls to the solver, with dynamic filtering.

---

**Input:** Set of demonstrations $\mathcal{T}$ with neurally inferred action and perception sequences

**Output:** Program summarizing the different demonstrations, or unsat

$\mathcal{F} \leftarrow \{\tau \in \mathcal{T} \mid \mathrm{actionconf}(\tau) \geq 0.98\}$        ▷ Static filtering by action confidence

$\tau_1, \ldots, \tau_k \leftarrow$ elements of $\mathcal{F}$ sorted by decreasing perception confidence

**for** prop $\in [1, 0.95, 0.9, 0.8, \ldots, 0.2, 0.1]$ **do**      ▷ Dynamic filtering by perception confidence

     $u \leftarrow \lceil \mathrm{prop} \cdot k \rceil$        ▷ Determines number of demonstrations

     $\mathcal{G} \leftarrow \{\tau_1, \ldots, \tau_u\}$        ▷ Demos with highest perception confidence

     **for** n $\in \mathrm{range}(\mathrm{max\_n})$ **do**      ▷ Progressively increase number of control-flow statements

         solution $\leftarrow \mathrm{solver}(\mathcal{G}, \mathrm{n})$        ▷ Call to Rosette solver

         **if** solution is not unsat **then return** solution

**return** unsat

---

Table 5: Ablation of heuristics on Karel.

| % | Karel | | |
|---|---|---|---|
| Model | Execution | Program | Sequence |
| demo2program | 72.1 | 48.9 | 41.0 |
| watch-reason-code | 74.7 | 51.2 | 43.3 |
| PLANS | $91.6 \pm 1.3$ | $53.9 \pm 1.0$ | $34.2 \pm 0.5$ |
| PLANS (- heuristic 1) | $91.8 \pm 1.4$ | $49.4 \pm 0.9$ | $30.4 \pm 0.5$ |
| PLANS (- heuristic 2) | $89.4 \pm 1.2$ | $49.2 \pm 0.9$ | $30.4 \pm 0.4$ |

## C.2 Solver heuristics

We use two additional heuristics to improve program and sequence accuracy on the Karel dataset:

1. Certain synthesis problems can be solved indifferently with an if or a while statement. We observed that choosing the solution using while yields better perfomance.

2. Some programs involving loops have several satisfying solutions, in which the size of the block before the loop differs. We observed that choosing the solution with the smallest number of actions outside the loop body improved program and sequence accuracy.

In order to measure the influence of these heuristics on performance, we made an ablation experiment. Results are reported in Table 5. We observe very little influence of the heuristics on execution accuracy. However, they yield around 4% absolute improvement on program and sequence accuracy.

## C.3 DSL encoding in Rosette

In Figure 6, we describe the encoding of the different control-flow constructs in the DSL. This encoding is for the Karel environment that has five different actions and five perception primitives. The ViZDoom encoding is exactly similar except that it has more actions and perception primitives. This encoding does not consider nested control-flow constructs: Indeed, we observed experimentally that these are not necessary to obtain satisfying accuracy. Besides, this allows for faster solver calls as this reduces the size of the search space. However, if this comes to be needed in other application domains, this assumption can easily be lifted by slightly modifying the encoding.

# D   Analysis of temporal complexity

In the section, we analyze the complexity of our algorithms. The static filtering algorithm makes $O(\text{max\_n})$ calls to the solver, where $\text{max\_n}$ is the maximum number of control-flow statements allowed in the generated program. The dynamic filtering algorithm makes $O(\text{max\_n} \cdot \text{n\_prop})$ calls to the solver, where $\text{n\_prop}$ is the number of iterations of the outer loop that increments the perception filtering threshold. In both cases, all solver calls are independent and can be performed in parallel. Therefore, the bottleneck of our algorithms is the duration of the longest solver call.

In both environments, we measured the duration of the longest solver call for each test program, and we averaged the measurements over all instances. For comparison purposes, we also report the average time taken by the neural component of PLANS to infer the specifications for one program. We performed all experiments on a machine running Ubuntu 18.04 with 2.00GHz Intel Xeon E5-2650 CPU and using a single GeForce RTX 2080 Ti GPU. The resulting values are reported in Table 6. We also report training time of the neural component.

In the ViZDoom environement, we observe that for a given program, the time spent by the neural component to infer the specifications and the longest solver call have same order of magnitude. This means that our model has no significant computational overhead with respect to the fully neural baselines. In the Karel environment, we observe that the duration of the longest solver call is one order of magnitude higher. In our experiments, we observe that the longest solver call is always the last one, with $n = 2$. If $\text{max\_n}$ is decreased, the average duration falls below $3s$, but at the cost of an execution accuracy decrease of a few %.

Table 6: Time measurements for PLANS. We report training and inference time. Training time corresponds to the whole training process. Inference time is measured for each program individually and averaged on the whole test set. For inference, we measure separately time spent inferring specifications with the neural component, and time of the longest solver call.

|  | Karel | ViZDoom |
|---|---|---|
| Training | ∼ 10 hours | ∼ 2 days |
| Inference of specifications | 1.39s | 3.68s |
| Longest solver call | 12.43s | 2.28s |

```
(define-synthax Action
  ([(Action previous_actions)
      (append previous_actions (list (choose 0 1 2 3 4)))]))

(define-synthax (Action_Block previous_actions length)
  #:base previous_actions
  #:else (choose
          previous_actions
          (let ([new_actions (Action_Block previous_actions (- length 1))])
            (Action new_actions))))

(define-synthax Positive_Condition
  ([(Positive_Condition previous_actions perceptions)
      (list-ref (list-ref perceptions (length previous_actions)) (choose 0 1 2 3 4))]))

(define-synthax Condition
  ([(Condition previous_actions perceptions)
      (choose
       (list-ref (list-ref perceptions (length previous_actions)) (choose 0 1 2 3 4))
       (not
        (list-ref (list-ref perceptions (length previous_actions)) (choose 0 1 2 3 4))))]))

(define-synthax If
  ([(If previous_actions perceptions)
          (if (Positive_Condition previous_actions perceptions)
              (Action_Block previous_actions 10)
              (Action_Block previous_actions 10))
          ]))

(define-synthax While
  ([(While previous_actions perceptions)
      (let loop ((aux1 previous_actions) (len 10) )
        (if (and (Condition aux1 perceptions) (>= len 0))
          (loop (Action_Block aux1 10) (- len 1))
          aux1))]))
```

Figure 6: Rosette encoding of the different DSL constructs for the Karel dataset.

## Footnotes

[1] `github.com/shaohua0116/demo2program`