[Reviews · NeurIPS 2020]

Review 1

Summary and Contributions: This work develops a method for synthesizing policies--represented as symbolic programs--from videos of demonstrations (sequences of states/actions). The idea is to train neural networks to map from raw perceptual input (videos paired with actions) to a symbolic state representation, and then use a symbolic solver to synthesize a policy (ie a program) which maps those symbolic states to the corresponding actions - thus supervising on symbolic state representations, but not on ground-truth programs. The technical challenge here is that the neural networks, acting as a perceptual front-end, sometimes make misclassifications. The symbolic solver cannot tolerate errors in the specification. Therefore, they propose discarding demonstrations where the neural network is uncertain in its outputs (ie, if the network is performing binary classification, whenever the network outputs a value close to 0.5 rather than close to {0.0,1.0}). They propose two different approaches for performing this discarding operation, and find that incrementally discarding the demonstrations with the most uncertainty yields the best results, as measured by both consistency with input/output examples and recovery rate of ground truth programs.

Strengths: The problem is important. From the point of view of program synthesis, their problem is important because inputs/outputs are not always the most natural specification language for end-users. From the point of view of imitation learning, it is important because policies represented as programs might generalize more strongly outside the training distribution, require less data to learn, and provide stronger guarantees or even admit formal verification. The paper should highlight these strengths in the introduction.

Weaknesses: From a technical point of view, one particular aspect of the approach is somewhat puzzling. The proposed method throws out entire input-output example traces if even a single action/state is uncertain, which seems excessive. Why not instead discard exactly those perception-primitives/actions which the neural models are uncertain about, and let the solver decide what the action/percept should be at that time? Such an alternative approach would be conceptually related to angelic programming (Barman 2010), and could lead to greater sample efficiency - Figure 3 suggest around 20-25 demonstrations are needed for successful learning, and perhaps this number could be brought down through judicious selection of which percepts to discard.

Correctness: I'm very happy with the evaluation methodology: they test both whether they can recover ground truth programs, programs semantically equivalent to ground truth programs (on all possible inputs), and programs observationally equivalent to ground truth programs on held out inputs. Other work in program synthesis should similarly evaluate on all three of these metrics, and it was refreshing to see this work do so.

Clarity: The paper is reasonably well-written, although it appears to omit definitions of the "perception primitives" and action primitives. These are very important because they define the symbolic abstraction of the MDP.

Relation to Prior Work: The relation to Ellis 2018 (which the authors discuss) should be reframed. They also learn to infer specifications from noisy perceptual input, which are then fed to a downstream symbolic solver, and also addresses the challenge of uncertainty over specifications, albeit in a Bayesian way rather than via the heuristics proposed here. Could you similarly situate your system in a probabilistic framework, and resolve the ambiguity over specs in a less heuristic manner? Would that fare better or worse on your data sets? I feel this is the main substantive difference, rather than the details which are presently emphasized in the text. The authors should put their heuristics front-and-center in this comparison. A critical missing comparison is to Chen 2019: Execution-guided neural program synthesis. It is essentially at ceiling for Karel. Is the reason why you don't compare because they supervise on programs, or because you go from raw perceptual input? If it's the former, you should include their results but specify that you use less supervision. Table 3 of the supplement suggests that your model, like that of Chen 2019, takes as input the Karel world pre-parsed into symbolic tokens. l159: also cite Solar-Lezama 2008 l178: have you considered an alternative scheme where you first ask the solver for any solution at all (not subject to any bound on total cost), then ask for a solution with a smaller cost, etc. until the problem is unsatisfiable? This would likely work better and has precedence in the literature (Singh 2012, as well as Cropper's MetaOpt system, see "Learning efficient logic programs" 2018). The the dichotomy of "rule-based" and "statistical learning" has some issues: l33: Gaunt 2016 frames inductive synthesis as a supervised learning problem, and so should not be contrasted with (Parisotto 2017) along this dimension. Furthermore (l34) Parisotto 2017 is "rule-based" in the sense that it (learns to) infer expressions in a user-defined domain specific language l39: in what sense does DeepCoder (Balog 2017) illustrate a serious scalability issue? If anything, this work demonstrates a scaling path through integration of neural and symbolic modules

Reproducibility: No

Additional Feedback: Post rebuttal: Thanks for the rebuttal. I'd encourage you to explore the angelic programming-esque version of your algorithm - to be clear, I'm not saying that this work is/would be obsoleted by angelic programming, simply that angelic programming is a helpful metaphor for explaining a version of your algorithm which does not discard entire execution traces. In no way does the comparison to angelic programming impinge upon the novelty of your work. Regarding "Progressively decreasing cost bound until problem is unsat": This alternative algorithm should have zero average effect on the final program. Instead, it is a scheme to more efficiently home in the minimum cost program. The data you give in the rebuttal is consistent with this (there is no significant changes in accuracy). The correct metric to measure here is the synthesis time, although what you did measure is a good sanity check that the accuracy remains essentially unchanged. Regarding comparisons to Chen 2019/Ellis 2018: Thanks for clarifying the differences in setup from Chen 2019. I think citing it is optional, but a good idea: on the one hand, they are the state-of-the-art, and you could in principle use a execution-aided synthesizer instead of a SMT solver. On the other hand I agree that their overall problem has a different API. I still disagree about Ellis 2018: see Section 4.1 of "Learning to Infer Graphics Programs from Hand-drawn Images" - correcting errors in the final neural network output does not depend on intermediate renders. The authors seem willing to make the requested changes (and in fact have already run an additional experiment, although as described above under "Progressively decreasing cost bound until problem is unsat" this experiment needs to measure solve time rather than accuracy). Contingent on these revisions I'm increasing my score to 6.


Review 2

Summary and Contributions: The paper presents a novel method in the field of program synthesis directly conditioned on high-dimensional video demonstrations. In this setting, the model has to predict the underlying program that produced a set of demonstration videos while being given access only to the video data. The proposed method, PLANS, is a multistage program synthesis procedure that first begins with a deep network predicting a set of perception primitives. These perception primitives perform the role of true/false predicate values in the domain specific language (DSL) that is being searched over to produce a program. Program synthesis is performed using an off-the-shelf rule-based program synthesizer, in this case Rosette, and several techniques are developed and ablated to perform synthesis even in the face of predicate misclassification. The experimental results seem strong, with improvements over more direct encoder-decoder-style neural synthesis models, and ablations test the various components of the pipeline.

Strengths: From my understanding, PLANS deviates from previous work in the field by taking a new perspective on the program from demonstration problem. While previous work attempted to predict programs directly using an encoder-decoder framework, PLANS instead more distinctly separates perceptual processing and logical reasoning by first predicting each observation's predicates and then relies on established program solvers to do the heavy lifting in searching the DSL for a matching program. This split proposed by PLANS seems quite intuitive and provides some obvious benefits over previous approaches as far I can tell: - program-level supervision is no longer necessary, meaning human demonstrations can be directly utilized without requiring an explicit program. - the supervision PLANS does use, the predicate classification, seems like a much easier form of data to acquire compared to viable programs, essentially being image attribute labels. Overall, I believe the paper presents a new strong method which foregoes the completely end-to-end training perspective of previous work in favor of a more structured approach which explicitly delineates deep learning and program search.

Weaknesses: I think that the benefits PLANS gains through the use of a program solver also imparts significant rigidity to the approach. This is reflected in the mostly heuristic technique developed to deal with misclassified perception primitives, where adaptive softmax score thresholds estimate network uncertainty. There are additional heuristics developed that deal with the particularities of the program solver, such as the iterative procedure to find a program with a specific number of control-flow statements, and it is not obvious how generalizable this is to other DSLs (i.e. if an agent designer requires a program cost function over multiple attributes rather than just control-flow statements, iterative search becomes computationally difficult). Additionally, I have concerns on how relevant this work is to the NeurIPS community, as the machine learning contributions seem rather minimal. Firstly, the deep learning used is a standard image classification network and the method to do uncertainty estimation is relatively heuristic.

Correctness: The claims, methods and empirical methodology seem correct.

Clarity: The paper is well-written and easy to follow.

Relation to Prior Work: The paper provides a sufficient overview of previous work. The distinction between the proposed PLANS and past work is clearly stated, which is the use of a predicate classifier and program synthesizer.

Reproducibility: Yes

Additional Feedback: I am not totally convinced of the use of softmax scores as a notion of uncertainty. Have the authors tried more standard forms of network uncertainty estimation using e.g. an ensemble of predicate classifiers? Post-Rebuttal: Thank you for addressing my concerns. Even with its mostly heuristic construction, given the method's strong empirical performance I maintain my vote for borderline acceptance.


Review 3

Summary and Contributions: The authors propose a two-step approach to synthesise a formal program specification from demonstration videos and subsequently find a program that adheres to the specifications extracted from the demonstration videos. The first component is a sequence-to-sequence network trained on demonstration videos. As supervision, corresponding perception and action sequences are provided. Given previously unseen demonstration videos, this network extracts action sequences from them. The second component is a symbolic program synthesis solver which is used to find programs from a DSL that explain the action sequences. To deal with noisy examples, the action sequences extracted from the videos are ordered by their confidence level and are pruned in least-to-most confidence order until a program can be found that explains all sequences.

Strengths: The authors tackle an interesting problem and the omission of ground truth programs during training is a realistic and useful assumption.

Weaknesses: W1 The paper relies heavily on empirical evaluation as there is little theoretical grounding of the method. However, the evaluation is not very thorough: only two datasets (and a variation) are used and only one internal component is evaluated (static vs dynamic thresholds). Why are thresholds used for ViZDoom but not for Karel? How dependent is the problem on a correct description of perception primitives? And how does it scale wrt, eg, perception primitives? W2 The omission of ground truth problems while training is interesting, however, the translation from action sequences to specification is not well explained. It seems that the translation is rather straightforward while phrasing the problem as "learning specifications" seems to imply a less narrow approach. As far as I understand the program learns action / perception sequences which can trivially be encoded as a program specification (please correct me if I am wrong). How restrictive is this formalism? W3 While the problem is interesting, the general problem setting is not novel and the methodology is the paper is quite straightforward and brings few novel technical contributions. The authors focus on the existing use case and benchmarks of video game analysis fails to provide a convincing argument that the change in setting (not requiring ground truth training programs) makes the problem (learning from video demonstrations) more general and significant.

Correctness: Few hard claims are made, therefore, generally they appear correct. Empirical methodology was already addressed in Weaknesses.

Clarity: The paper is well written and rather accessible.

Relation to Prior Work: Relevant related work and the broader context are well summarised. It is clear how this work differs from previous contributions.

Reproducibility: Yes

Additional Feedback: Post rebuttal: Thank you for your clarifications. I still recommend this paper to be accepted, however, I did not upgrade my score because while I understand the paper better now, I feel that some of the weaknesses remain.


Review 4

Summary and Contributions: The paper proposes PLANS, a framework for inferring programs from demonstration videos. The framework consists of two stages: (1) it uses a neural architecture to extract information about states and actions from videos, and then (2) it uses a rule-based system to synthesis a program that maps from observations about the states to actions. The experiment shows that it outperforms existing approaches on two standard benchmarks in terms of accuracy. Besides the empirical result, the main technical contributions are: 1. A new framework for learning programs from videos using combined neural architecture and rule-based systems; 2. A confidence-based heuristic to filter noises in the input to rule-based synthesizer.

Strengths: 1. A novel framework for combining neural architectures and rule-based systems for the task of generating programs from video demonstration. This contribution can trigger interesting discussions inside the community given that there is a growing interest in combining neural approaches and symbolic approaches. 2. The empirical evaluation shows the approach outperforms the existing approaches by a respectful margin. 3. The paper is clearly written.

Weaknesses: 1. It is unclear why the proposed method to handle noise is better than the previous method (Raychev et al. 2016). The paper claims that the previous method only guarantees to find a program that is close to the correct solution. However, the proposed method has no guarantee that it will find the correct program. In particular, there is no guarantee that the confidence indicated by the neural nets is accurate. Some empirical justification is needed to support the claim the the proposed method of dealing noise is better. 2. It is unclear how the proposed method scales to larger programs. Rule-based synthesis systems, especially SAT/SMT-based systems, are know to have difficulties scaling to large programs. 3. The high-level idea of combining neural architectures and rule-based systems to learn from visual inputs is not new (e.g., Ellis et al., 2015). However, I do agree that the setting and the application are new.

Correctness: Yes. The paper doesn't make any strong theoretical claim. And the experiment methods are right to me.

Clarity: Yes. I could follow the paper easily even though the specific problem is new to me.

Relation to Prior Work: Yes, to my knowledge.

Reproducibility: Yes

Additional Feedback: What are the sizes of the generated programs? How do you see the proposed approach scales to larger programs? After rebuttal: Thanks for the thoughtful response. I remain positive.

[Author Response · NeurIPS 2020]

**All reviewers - Comparison with Ellis 2018.** A crucial component of Ellis 2018 is the ability to incrementally render
specifications inferred from the input image and measure similarity with that image. This similarity is used as a
Bayesian likelihood for specifications sampling. However, this is not applicable to our setting as there is no direct
equivalent to the incremental rendering process. Our approach offers more generality for neuro-symbolic systems.
**Action/perception primitives.** In Section 2.2, we opted for an abstract definition of action and perception primitives as
these are domain dependent. For instance, Vizdoom actions include "move forward", "turn left" and "attack". Vizdoom
perceptions include "is there a monster?" or "is a revenant in target?". We will add a detailed list in the paper.

**Reviewer 1 - Discarding single tokens instead of entire traces.** This is a promising direction for future work to
improve sample efficiency. Yet there is one fundamental difference with Barman 2010. Angelic programming allows
for uncertainty in the program, while the approach proposed by the reviewer introduces uncertainty in the specifications.
We agree that there is a conceptual connection, but it is not obvious to us whether both problems are equivalent.
**Comparison to Chen 2019: Execution-guided neural program synthesis.** We do not compare because the synthesis
tasks are different. To cite Chen 2019 (about demo2program) "Sun 2018 propose to synthesize the program from
demonstration videos, which can be viewed as sequences of states. In such a problem, all intermediate states can be
extracted from the videos. On the contrary, in the input-output program synthesis problem studied in our work, the input
to the program synthesizer provides only the initial state and the final state." Chen 2019 infers likely intermediate states
in order to simplify synthesis, while in our case they are part of the specification. This makes the tasks incomparable.
**Related work.** We will refer to Solar-Lezama 2008 for the concept of program sketching, and improve the dichotomy
between "rule-based" and "statistical" methods. We will correct the reference to DeepCoder.
**Progressively decrease cost bound until problem is unsat.** We have implemented an alternative decremental scheme,
preliminary results do not show a noticeable performance gap. We thank the reviewer for the literature references.

| Accuracy (%) | Karel | | | ViZDoom | | |
|---|---|---|---|---|---|---|
| PLANS (best configuration) | 91.6 | 53.9 | 34.2 | 88.0 | 65.5 | 58.8 |
| Decrement cost bound (best configuration) | 92.1 | 53.8 | 34.5 | 87.5 | 64.8 | 57.8 |

**Reviewer 2 - Rigidity of the approach.** The filtering technique developed to deal with missclassified perceptions
primitives does not make domain specific assumptions and is potentially applicable to similar neuro-symbolic systems.
**Cost of iterative search over multiple attributes.** A possible solution to this problem is to directly encode the cost
function over programs in the solver. This is not possible in Rosette, that we chose for its flexibility in encoding DSLs.
Future work could attempt to build on different solvers integrating cost functions and compare performance.
**Impact to the NeurIPS community.** We believe that developing efficient neuro-symbolic systems is crucial to the
machine learning community. Dealing with specification uncertainty is one of the major challenges involved.
**Softmax scores vs. ensemble methods.** Our method is orthogonal to the choice of uncertainty measure. Ensemble tech-
niques provide a good measure of uncertainty, but there is a high computational cost involved with training sufficiently
many members. This is why we followed prior work using softmax response, which yields good performance.

**Reviewer 3 - No thresholds for Karel.** There is no need for filtering as the network reliably infers the specifications.
**Perception primitives.** No semantic understanding of the perception primitives is required for our method. Besides,
our filtering technique introduces some resistance to noisy training labels as a side effect. Concerning scalability, our
empirical observation is that program length and control-flow depth matter more than number of primitives.
**Encoding of actions/perceptions into I/O specifications.** High-level semantics of the encoding are indeed intuitive.
The technical challenge lies in implementing these semantics accurately in the Rosette solver based on Racket.
**Technical contributions.** The ability to deal with uncertainty in specifications has been repeatedly described as a
crucial aspect of neuro-symbolic systems, since traditional symbolic solvers are extremely vulnerable to noise. We
addressed the key technical challenge of dealing with misspecifications in a generic way.
**Generality of the action/perception framework.** It is a very natural way of describing agent-environment interaction.
An important direction for future work is to generalize to arbitrary combination of perceptions in conditionals.
**Advantages of removing program supervision.** (i) action/perception labels are much easier to acquire. (ii) it removes
the need to train on several different demonstrations of the same program . (iii) it drastically reduces training cost.

**Reviewer 4 - Experimental comparison with Raychev 2016.** This is problematic, because the algorithm in Raychev
2016 deeply modifies the traditional program synthesis process. It was only evaluated on bit-stream programs, and has
not been integrated in a general purpose solver yet. In contrast, our filtering algorithms build on top of state-of-the-art
solvers, and are simple to implement. We have contacted the authors of Raychev 2016 and agree on additional
advantages of our approach in a neuro-symbolic setting (i) it focuses on satisfying the most confident samples, which
makes finding the correct program more likely. (ii) sorting the samples by confidence reduces overall time complexity.
**Program size - scalabilty.** Programs are composed of up to 43 tokens, and we observe balanced time measurements
between neural network inference and symbolic solver calls. We believe that parallelism inside solver calls could help
scaling to larger programs with more control-flow statements. This would be fair as the network runs on a GPU.

[Meta-Review · NeurIPS 2020]

This paper presents a method for synthesizing symbolic policies from videos of demonstrations. It learns to map from the video frames to a set of predicates. A second stage program synthesis algorithm then searches for a program in a DSL that is consistent with the sequence of predicates. There is mostly a consensus that the problem setting is interesting, and the approach is a sensible-but-heuristic piecing together of a standard neural component with a standard program synthesis component. Thus, the strength of the paper is not in the methodology, and there were some concerns about whether this kind of contribution is sufficient for NeurIPS. However, the overall problem framing is interesting in its ability to go all the way from perceptual input (video demonstration) to symbolic program representations. On the strength of this as an interesting conceptual approach and the fact that the paper successfully executes on the approach, it seems worth accepting.